# What Are the Challenges Faced by Village Doctors in Provision of Basic Public Health Services in Shandong, China? A Qualitative Study

**DOI:** 10.3390/ijerph16142519

**Published:** 2019-07-15

**Authors:** Qian Wang, Yuejia Kong, Jiyao Sun, Yue Zhang, Linlin Yuan, Jian Wang

**Affiliations:** 1School of Health Care Management, Shandong University, 44 Wenhuaxi Road, Li Xia District, Jinan 250012, China; 2NHC Key Laboratory of Health Economics and Policy Research, Shandong University, Jinan 250012, China

**Keywords:** basic public health services, village doctors, challenges, China

## Abstract

*Background*: Village doctors, as gatekeepers for the health of rural residents in China, are confronted with adversity in providing the basic public health services (BPHS), which has significantly impeded them from providing high quality BPHS. This study aimed to explore the obstacles and difficulties faced by village doctors in order to improve the quality and efficiency of BPHS provision and increase the health level of the population. *Methods*: In-depth interviews were employed to conduct this qualitative study. A total of 51 village doctors in four cities of Shandong Province were interviewed. The interviews were transcribed, anonymized, and imported into NVivo11.0 to facilitate management. Thematic framework analysis employing the constant comparison method was applied to the data analysis. *Results*: The main challenges faced by village doctors comprised the shortage, gender imbalance, and poor education of village doctors; older village doctors in some villages; low income; lack of social security; inappropriate performance assessment; inadequate professional BPHS training; heavy workload; and insufficient cooperation from rural residents, which have exacerbated the quality, efficiency, and accessibility of BPHS to some extent. *Conclusions*: Village doctors, as the important BPHS providers in rural Shandong, are facing a wide range of challenges. It is urgent for government officials and policy makers to consider these challenges and concentrate on improving the quality of BPHS provision by developing relevant and practical strategies.

## 1. Background

Providing public health services has a great significance in continuously raising the awareness of health for all residents [1], effectively preventing and controlling the occurrence and prevalence of major infectious diseases and chronic diseases [2], promoting the equalization of basic public health services (BPHS) in urban and rural areas [3,4], and improving residents’ health literacy and quality of life [5]. Since the New Health System Reform launched in 2009, the provision of basic public health services (BPHS) in rural China has been propelled and revitalized by emphasizing the function of village doctors [6]. In October 2016, the State Council of China promulgated the plan of *“Healthy China 2030”*, which also seized the key point that the village doctors, as gatekeepers for the health of rural residents, are taking indispensable responsibility for providing BPHS with high quality, efficiency and accessibility in rural China, and gradually promoting the realization of BPHS equalization [7]. In 2017, the BPHS package has increased to 12 categories (Appendix A
Table A1).

In rural China, BPHS has been provided through a three-tiered system of county hospital and center for disease control and prevention (CDC), township healthcare centers (THCs), and village clinics [8], where village doctors, rooting in rural residents, are serving the bottom-tier and definitely on the front lines of BPHS provision [9]. Village doctors, formerly known as “barefoot doctors”, serve as the bottom-tier of the Chinese rural health system [6]. Barefoot doctors are farmers who work in rural China through elementary basic medical training. In the early 1980s, when China implemented opening up policy and economic reforms, barefoot doctors were unable to meet the growing demand for health services of rural population. In 1985, the Chinese government stopped using the term “barefoot doctor” and replaced it with “village doctor” [10]. Barefoot doctors need to pass an exam to get a village doctor’s certificate. Those who fail will quit the health service industry [11]. The “Regulations on the Administration of Village Doctors” implemented since 1 January 2004 stipulates that village doctors, after passing the corresponding registration and training examination and obtaining the practice certificate, can then start a business with a formal license [12]. Village doctors are encouraged to apply for the standardized certificate like a practicing physician certificate. The initial training level for village doctors was only a few months, but now they must receive two to three years of professional training, which is equivalent to a high school diploma.

The data of the latest Sixth National Population Census of China showed that nearly half of population (42.65%) are living in rural areas [13], which demonstrated that supplying high quality BPHS in rural China is of great significance in improving the population’s quality of life and promoting social harmony [14]. Currently, about 1 million village doctors undertake approximately 40% of the BPHS workload, which also lays stress on village doctors’ significant function in delivering BPHS to rural residents, so the quality of BPHS provided by village doctors has a direct influence on the health status of rural residents [15]. However, a few studies have uncovered that village doctors are facing many challenges in BPHS provision, mainly including heavy workload, low income, lack of social security, and insufficient cooperation from rural residents [15], which have heavily worsened the quality of BPHS provision for rural residents [9].

Some studies have focused on BPHS in recent years. A survey conducted in South Sudan explored spatial accessibility to BPHS among counties and suggested that those counties with the poorest health access should be targeted for priority expansion of clinical services, while no attention was paid to village doctors [16]. Zhao’s [14] study focused on the barriers to the implementation of measures to improve BPHS in the Chinese urban community instead of village clinics in rural areas. Another published study [17] explored influence on the performance of village doctors in the provision of BPHS before and after they adopted performance-related contracts enacted by the Chinese government, while no aspect presented in the article reflected barriers faced by them when delivering BPHS. On the other hand, some published studies have focused on basic medical service provision of village doctors, such as prescription behavior and potentially unnecessary care, paying less attention to BPHS [18,19,20]. Xu’s [21] longitudinal survey and Li’s [9] research identified the challenges and influencing factors of high quality BPHS provision by village doctors using quantitative questionnaire survey, such as aging, gender imbalance, low education, lack of social security and training opportunities, low subsidy, and so on, while they failed to conduct an in-depth interview to provide detailed qualitative information and reasons why village doctors are facing these challenges. Although Zhang’s [15] and Ding’s [22] research respectively uncovered the obstacles faced by village doctors in providing BPHS by performing a qualitative field survey in western China (Guizhou Province) and Central China (Hubei Province and Jiangxi Province), because of the difference in socio-economic development and demographic characteristics, the evidence observed in these studies may not totally generalize to the context of village doctors in rural Shandong [23]. Therefore, studies on this subject are still lacking in Shandong, which has the largest number of village doctors (113,465) in 2016, ranking first in the country [22]. It is worth noting that Shandong has a large rural population ranking second in China [24], so investigation and analysis on challenges faced by village doctors in BPHS provision in rural Shandong could be more practical and more meaningful in facilitating the quality, efficiency, and accessibility of BPHS and guarding health for rural residents by providing relevant reference.

The aim of the present study was to explore the challenges faced by village doctors in provision of BPHS in Shandong Province. A qualitative study design was used to provide abundant and in-depth information about the complex phenomenon of what is impeding village doctors from providing higher BPHS. Developing a well-rounded understanding of challenges derived from village doctors’ perspectives and experiences in BPHS provision could inform the development of practical BPHS-related polices and strategies, which could be able to facilitate the improvement of BPHS in rural China. Further, this analysis may have policy implications for other low and middle income countries who have difficulties in providing primary health services to achieve universal health coverage (UHC).

## 2. Methods

### 2.1. Study Areas and Subjects

This study was carried out in Shandong province located in the eastern region of China, which is the second largest province by population and third largest by economy in China. A purposive sampling was employed in this study. First, 4 of the 17 cities in Shandong Province were selected to represent different geographic regions and economic areas of the province: Qingdao, Zibo, Liaocheng, and Heze. These cities (in the above order) are located in the east coastal, central, western, and southwest regions of Shandong Province and ranked second (119,215 yuan), third (101,569 yuan), fifteenth (49,806 yuan), and seventeenth (32,558 yuan), respectively, by per capita gross domestic product (GDP) according to 2017 statistics [25].

Second, within each city, three counties were then chosen by the local Center for Disease Control and Prevention (CDC) to represent counties with various level of BPHS activities. Next, three representative township health centers (THCs) were chosen from each county. Finally, within each THC, two village clinics with good and bad BPHS work were then recommended by each THC to be included in the final stage of the sampling process. In total, 72 village clinics were initially included. Within each village clinic, one village doctor was designated as the interviewee for the clinic. Village doctors refer to those who provide basic medical service and BPHS in village clinics, except nurses and other medical professionals. The interviewee first completed a self-administered paper questionnaire that collected basic personnel information of all village doctors in his/her clinic. Then, after obtaining oral consent, a face-to-face semi-structured in-depth interview with the interviewee was conducted. Because of scheduling conflicts and considering the data saturation [26] in data collection and analysis, we were finally able to interview 51 village doctors, and gathered 129 village doctors’ basic personnel information from 61 village clinics.

### 2.2. Data Collection

A paper questionnaire was first administered to the designated interviewee in each clinic to collect the staffing information and basic demographics about all village doctors serving in the clinic (the number of doctors, their gender, age, educational background, specialty, qualification, years of working as a village doctor, and so on). After obtaining oral consent, an in-depth interview with the interviewee was then conducted. During the interview, investigators who were uniformly trained conducted 40–60 min face-to-face in-depth interviews with the respondents based on the pre-designed semi-structured interview outlines. The guidelines focused on challenges that have hindered village doctors from providing high-quality BPHS in various aspects including manpower situation, doctor’s age, gender, income, education, qualification, health insurance coverage, workload, THCs’ supervision, financial incentives, professional BPHS training, as well as villagers’ cooperation to services and solicited suggestions or identification of additional problems. During the interviews, body language, such as facial expression, tone, and small movements of the interviewees, was carefully observed and noted, so as to help judge the authenticity of the interviewee’s words. If there is any problem, the interviewer will continue to ask questions to collect more authentic information. The interviews were conducted either in the meeting rooms of the THCs or the village clinics, and were audio-recorded with the interviewee’s permission. Subsequently, the interviews were transcribed verbatim in Chinese by a different person outside the research team [27]. All Chinese transcripts were checked by authors Wang Qian and Sun Jiyao so as to minimize data loss. All interviews were completed in August 2017.

### 2.3. Data Analysis

Two sources of data were collected in this study: basic information of village clinics staffing and in-depth interviews. The basic information of staffing provided overall demographic situation of village doctors, which helped to better understand the distribution of village doctors with different characteristics and offer reference to the further discussion of challenges faced by the village doctors in providing BPHS. For qualitative data collected from in-depth interviews, principles and procedures of thematic analysis employing the constant comparative method instructed data analysis [27,28,29], following transcription and entry into the qualitative analysis computer program Nvivo11.0 (QSR international, Melbourne, Australia) [30]. Concurrent data collection and data analysis occurred with codes and categories being inductively developed from the data. The analysis involved identifying codes and their properties and dimensions, grouping these codes to create categories, systematically comparing and contrasting the codes and examining the connections between the categories and subcategories [27].

To be specific, first, a preliminary analysis of the original data was conducted by Qian Wang and Jiyao Sun using Nvivo11.0 to identify the initial coding, which was carried out in Chinese, the native language, in order to avert misunderstanding and to minimize the risk of losing participants’ original meanings [27]. After that, the properties and dimensions of initial coding were compared, analyzed, and grouped to create categories. Subsequently, Qian Wang and Jiyao Sun continued to systematically compare and classify the data segments and tried to explore and identify the general categories based on the discovered subcategories. At each step, the consistency of coding and categorizing was checked by authors. Any differences found in each process were discussed among the author group until agreement was reached. Next, data related to emerging themes were translated into English, and the translated versions and the original Chinese versions were checked by an independent and bilingual person outside the research team in order to link the relationships between categories on a conceptual level rather than on a descriptive level [31].

### 2.4. Ethics Approval

Ethics approval for this study was obtained from School of Health Care Management, Shandong University in China (protocol number ECSHCMSDU20190303). Pseudonyms have been used to preserve participants’ anonymity.

## 3. Results

### 3.1. Village Clinic Staffing Distribution

A total of 61 village doctors completed the questionnaire about their clinics, reporting the basic information of 129 village doctors who provide health services in the village clinics of these village doctors (Table 1). A total of 28 village clinics had only one village doctor on duty, and each served approximately 232–1800 local rural residents. Also, 12 village clinics had 2 village doctors and served about 350–1842 residents. The remaining 21 clinics had 3–5 primary healthcare providers, serving 1023 to 6012 local residents per clinic. Table 1 showed the majority of village doctors and health providers were aged 36–55 years old, and male village doctors occupied almost two-thirds of them. For education and specialty, village doctors who had a secondary school education and gained a specialty of western accounted for the vast majority, 76.74% and 57.36%, respectively. Those who acquired the qualification of Rural Doctor constituted 77.52%, while a mere 10.08% of village doctors went to Practicing Physician. Most of the respondents (44.19%) worked as village doctors for 21 years or more.

### 3.2. Key Challenges in BPHS Provision

A total of 51 village doctors consented and participated in the in-depth interview. The key challenges derived from village doctors’ perspectives and experiences in BPHS provision were discussed below.

#### 3.2.1. Shortage of Village Doctors

According to the interview, shortage of village doctors was identified as one of the main barriers. This study revealed the same situation in 28 villages (54.90%), where only one village doctor serves the whole village, and seven of these villages had a population of more than 1000, or even 1800. For example, in village C1, one village doctor was responsible for providing BPHS for 1037 villagers, and even worse, in village A6, only one village doctor took the responsibility of BPHS delivery, which covered 1687 villagers.
“I am the only doctor of the village clinic who is responsible for both routine basic medical diagnosis and treatment and BPHS. The workload is heavy, the task is difficult, and the pressure is increasing.”*(A9)*
“There are 823 villagers in my village. Actually the health manpower is not enough. For example, it always takes me a long time to establish rural residents’ electronic health records by inputting large mounts of data on computer. I really need someone to help me share the work task.”*(C23)*
“I always need to walk around to provide public health care services. Because I am the only doctor in the clinic, I must shut down the clinic when I am off to provide BPHS. Villagers often complain that they can’t find a doctor when they need to check some diseases.”*(D13)*

#### 3.2.2. Older Village Doctors in Some Villages

Age of village doctors in some villages may be another challenge to BPHS. In some of the clinics, village doctors are older, and the service population is large, which greatly reduced the quality of service. For example, in village C3, there are two village doctors in the clinic, one 55 years old and one 58 years old, serving 1609 villagers. There is even village B1, where population is 1800, and there is only one village doctor who is 57 years old in the village clinic. Older age has become another major obstacle for village doctors in providing high quality BPHS. Older village doctors often needed more time and energy to learn and comprehend new knowledge about how to provide better BPHS. It is also difficult for them to develop a good command of new skills that are necessary for delivering BPHS, such as computer proficiency due to the deterioration of physical function and memory. Another worrying situation is that some older villages faced the problem of having no successor after retirement.
“I’m getting older and I not familiar with the operation of new electronic products. I rely on my children to input health-related data into computer to establish residents’ health records, because it is pretty laborious for us and my computer skills are really poor.”*(C19)*
“My knowledge is also outdated and I always don’t understand those new policies and strategies issued by central government on how to conduct high-quality basic public health service.”*(C21)*
“I am over the age of 50, I don’t have enough physical strength and energy to carry out BPHS. Young people are not willing to be village doctors. I am very worried about who will continue to provide basic public health services when I retire?”*(C3)*

#### 3.2.3. Gender Imbalance of Village Doctors

Evidence found in this study showed that male village doctors account for the majority of doctors (64.34%). In village clinics with only one doctor, male doctors were even more prevalent (78.57%). Some specific BPHS aimed at females may be difficult and inconvenient to carry out if only male village doctors are on duty, so such services may be deliberately avoided.
“Since I am the only male village doctor here, it is very inconvenient to provide maternal and gynecological services for female rural residents… Unluckily, feudal and conservative customs and traditions often prevented women from seeking these services from male village doctors, so there are few such services conducted here.”*(A15)*

#### 3.2.4. Insufficient Education and Qualification

Among the surveyed village doctors, the majority (76.74%) only had secondary school education, and those with a college degree or above were rare. Some village doctors admitted that poor education has limited the delivery of BPHS to a certain extent.
“I only got secondary school education, the knowledge and skills I acquired are insufficient for me to provide high-quality public health services.”*(C15)*

Although 77.52% of the village doctors had the qualification of Rural Doctor issued by local government, by comparison, doctors who gained credentials of Practicing Physician and Practicing Assistant Physician certificated by central government, which were more authoritative, only accounted for 10.08% and 10.85%, respectively. Furthermore, more than a half of the qualified village doctors gained a specialty of Western medicine (57.36%), and nearly one-third of them were integrated traditional and Western medicine doctors (28.68%), which illustrated that rural areas still lack professionals who gain educational background of preventive medicine and public health; to some extent, this hinders the quality of BPHS provision.
“I have limited professional ability, and I am not skilled. It’s always difficult for me to carry out some BPHS-related work.”*(B7)*
“For some part of basic public health services, we couldn’t even gain a deep understanding, let along provide satisfying services for rural residents”*(D9)*

The village doctors also expressed their longings to improve professional skills through education.
“I recommend that the superior department, like township health centers and centers for disease control and prevention can assist us and conduct more training related to basic public health services.”*(D16)*
“I hope to be trained by a professional public health agency so that I can deepen my understanding instead of working in an amateur way, as it is now.”*(D9)*

#### 3.2.5. Low Income and Lack of Social Security

Subsidies allocated to village doctors could directly influence the provision of BPHS in rural China over the long-term [9]. Currently, there are three main sources of income for village doctors: essential drug subsidies, BPHS subsidies, and profitable basic medical service fee (diagnosis fees and injection fees), with BPHS subsidies accounting for the majority of the income [32]. Although most village doctors indicated that BPHS subsidies were issued in time, they expressed discontent about the amount of subsidies being inadequate and disproportional to workload. Village doctors now had to take up an increasingly heavier workload because of some new items and content added to BPHS compared with that before, which needed to devote more time and energy, and they do not think the current subsidy is worth their effort. Besides, providing BPHS has also largely influenced their income by carrying out profitable basic medical service (earning diagnosis fees and injection fees); as a result, the total income has decreased compared with that of previous years. Many village doctors even thought that their income was lower than that of migrant workers in the same village.
“This year, the content of BPHS has been enriched. According to the requirements of our superiors, we have to spend more time and energy on these services. Before the implementation of these services, we focus on profitable basic medical services, and we used to earn more money than those who worked outside the village, but now the situation is opposite, so I might as well go out to work.”*(A12)*
“Too little income, too few subsidies! We are not afraid of tired work, but now is the BPHS workload is too heavy, my time and energy spending are not proportional to income”*(D16)*

Village doctors also complained about the lack of social security, such as pension insurance and health insurance, also adding financial pressure to their face.
“Although Township health centers and village clinics were integrated in this area, we are still responsible for our own profits and losses. We still need to buy our own pension insurance and health insurance, and the financial pressure is high.”*(C5)*
“Health professionals working at township health centers and above had better government provided pension plans available to them, while we are not treated similarly. We are treated almost the same as the farmers! I felt disrespected and unfair’’*(C10)*

For the problems above, some village doctors put forward their ideas.
“I think we deserve the same pension plan as those who work in THCs. I hope government can incorporate village doctor’s social security into finical budget.”*(B12)*
“I hope I can have a steady basic salary as a living support regularly paid by government finance, and in a long term, the basic salary can be issued base on village doctors’ qualification, and working years, just like teachers in our country.”*(A1)*

#### 3.2.6. Unreasonable Performance Assessment on Village Doctors

THCs supervise and assess the performance of BPHS provided by each village clinic using a scoring system. The actual amount of BPHS subsidies distributed to village doctors was dependent on or varied with the assessment results of THCs. THCs had clear assessment criteria for village doctors’ BPHS work, mainly including telephone interview to villagers, and on-site inspection in clinics in order to confirm if malfeasance and fraud exist. However, several unreasonable facets should be considered during the performance assessment.

First, although the scoring system stimulated village doctors’ motivation to some extent, some village doctors complained that there was little differentiation of the distributed BPHS subsidies to them based on the assessment score, which means that whether a village doctor provides good or bad BPHS, they get almost the same amount of subsidies, damping their enthusiasm to achieve a higher score.
“The subsidy is issued according to assessment score. However, actually all village doctors have similar amount of subsidy regardless of the score is high or low in this area. That number (assessment score) doesn’t reflect and distinguish our performance”*(C12)*

Second, most village doctors voiced their dissatisfaction about the assessment not taking account of specific local situations, especially the telephone return visit to villagers. Besides, lacking feedback mechanism from THCs after performance assessment has frustrated village doctors’ initiative and removed their opportunities for improving the quality of BPHS based on the assessment feedback.
“For one thing, elderly people left their son’s or daughter’s phone number to THCs rather than their own numbers. When inspectors from THCs called back to check the implementation of BPHS, some young people didn’t know whether their parents received BPHS or not. For another thing, many elderly people have hearing problem, so they may miss the call, but the THC inspectors don’t check this situation and give me a low score. And sometimes, the questions that inspectors asked were so blunter and professional that some rural residents who are illiterate or old couldn’t even understand. All of these situation above will affect the assessment to us, and I’m going to get criticism from THC managers. This is very inappropriate and unfair.”*(D7)*
“After the assessment, there was no feedback from THCs. I don’t know what I should do to improve my services. Sometimes I feel that I did a good job, but after the assessment I found my score was deducted. Also, the deduction criteria are not transparent.”*(C5)*

In addition, some village doctors also complained about the high-frequency performance assessment from THCs, which has forced them to attach more importance to the quantity of services and neglect the quality in order to fulfill the assigned onerous BPHS tasks in time.
“The telephone spot checks of township health centers are so frequent. We sometimes don’t have enough time to finish all the work at all. In order to cope with the assessment, we had to finish the task as soon as possible and meet the quantitative requirement, while the quality of service was not guaranteed.”*(A1)*

Some advice was also presented by village doctors during the interview.
“The assessment mechanism needs to be further optimized, and the assessment criteria should be flexible. For example, performance assessment should give full play to the role of financial incentives. Village doctors who complete basic public health services with high quality should get more revenue…Special situation like old people miss the telephone return visit, should be considered……The frequency of the assessment should be determined according to the workload. Hope to give us enough time to finish the task with high quality, then we can also have a sense of achievement.”*(C11)*
“I hope that the assessment results can be publicized so that everyone knows how many points they get, and why their points were deducted.”*(D3)*

#### 3.2.7. Inadequate BPHS Training

At present, BPHS training for village doctors is mainly organized by THCs, rather than county-level CDC. THCs provide targeted training according to the BPHS package. However, although some village doctors mentioned that the training had a good effect on the improvement of their service capability, most village doctors still felt inadequate in their ability to provide high-quality BPHS after the training. First, even some THC staff who trained village doctors could not understand the contents of BPHS very well without help from superior department such as CDC, so the quality of BPHS training delivery by THCs was discounted.
“Basically, we have no connection with the county-level CDC. Almost all of the works were under the guidance of THCs. However, sometimes, People in THCs are not very good at some basic public health projects, and skill training and guidance to us are not professional enough. They only know how to assess our performance and deducted scores in parrot ways according to the assessment criteria, so what we learn is really limited”*(C11)*

In addition, in some surveyed place, there are rare trainings especially aimed at new government policies guiding BPHS implementation, so many villager doctors feel disconnected with relevant policies on how to provide high-quality BPHS.
“The training for village doctors is not enough. Some of the latest information and policy cannot be received and understood in time, so some services will not be carried out.”*(D6)*

Besides, some THCs replace BPHS training with regular working conferences, which dramatically reduces the training effect and contributes to more unprofessional service delivery.
“Currently, the BPHS training has been replaced by regular working meetings, and only a small part content related to BPHS was mentioned in the meetings. The superior department does not pay enough attention to the training and usually don’t care about the quality of the training.”*(D9)*

Village doctors also expressed the hope to strengthen BPHS training.
“I hope the county CDC can really take the responsibility to strengthen, facilitate and ameliorate BPHS training for primary level BPHS personnel, like us…County CDC and THCs should cooperate well, which means CDC should strengthen the training toward THCs, so that village doctors can be better trained by THCs.”*(D16)*

#### 3.2.8. Heavy BPHS Workload

Village doctors not only provided basic medical services, but also offered BPHS to the entire village. The BPHS package has been enriched from 22 items in 9 categories in 2009 to 49 items in 12 categories in 2017 [33,34]. Village doctors generally complained about BPHS tasks has been increasing, such as creating more detailed electronic health records on the computer, conducting follow-up services for key rural residents, increasing the frequency of health education, which were pretty laborious for consuming too much time and energy; and engendered negative emotions and decrease in initiative and motivation. Village doctors even shoulder a heavier workload if the village had a large population.
“Creating rural residents’ electronic health records on the computer is very burdensome. Villagers’ health records, you know, contain a lot of information, such as demographic information, physical examination information, disease-screening information, vaccination information and so on. It will take me a large amount of time to collect and input the information on computer for the whole village residents. What’s worse is that some rural residents even refuse to create health records for them, as they think these files are useless! Then I need to persuade them, I am quite tired everyday!”*(A5)*
“We need to work around to conduct follow-up services for six key groups of villagers including people suffered from hypertension, diabetes, severe mental disease, children, maternal, and the elderly, so this job is pretty intensive… I sometimes cannot find the person due to some reasons, like he/she is working outside when I come to his/her house to carry out follow-up services, and then I have to leave without any outcome. It’s so exhausting!”*(A13)*
“Too much workloads now! Compared to government officials and teachers who only need to work 8 h everyday from Monday to Friday, village doctors have to work 365 days a year without vacation. I conduct physical examinations and follow-up services during daytime, and I also have to stay up late to input health-related data on computer at night. I can’t get enough rest and sleep!”*(B3)*

Some village doctors also admitted that in order to complete the assigned BPHS work in time and cope with the high-frequency performance assessment from THCs, they only focus on the quantity of BPHS delivery, regardless of the quality.
“I have no more time and energy to care about and guarantee the quality of BPHS because too much work and multifarious trifles needs to be done, and I must finish the BPHS work in time according to the regulations of THCs. Take health education as an example, in order to save time and simplify the procedure, the health education always carry out without depth and pertinence, and the form was single and monotonous, so it’s obvious that the effect was not good.”*(A8)*

#### 3.2.9. Insufficient Cooperation from Villagers

Most village doctors mentioned the problem of non-cooperation of villagers, which hindered the implementation of BPHS. For example, some villagers are reluctant to receive the follow-up service and health-related management services, and some even refuse to see village doctors. Several reasons might account for this phenomenon.

First, villagers who have poor education or low health awareness could not fully understand and appreciate the significance of BPHS, because they always hold some wrong thoughts, like, among others, “I will be involved in discrimination from villagers if I am diagnosed some disease through physical examination”, “No symptoms mean I’m healthy. It’s alright for me without BPHS-related preventive care”, and “It’s okay and everything goes well without BPHS check. In case I check out some diseases, it is nothing but a psychological and financial burden on me”.
“It is difficult to manage patients with severe mental illness, tuberculosis and other infectious diseases, because they believe that these diseases may give rise to discrimination from other villagers, and damage the image and reputation of their family. Therefore, some family does not want others to know that someone in his/her family has such above diseases, just like a Chinese proverb says: “It is an ill bird that fouls its own nest.”*(D1)*
“Many villagers with poor health education believed that BPHS was unnecessary for them, which made the BPHS difficult to be carried out.”*(A5)*
“Some villagers had a heavy psychological burden about knowing they have some kinds of severe diseases, so they tried to evade seeing village doctors and receiving BPHS.”*(C6)*

Second, with BPHS, as a kind of preventive healthcare service, many villagers might not feel the rapid benefits and effectiveness to improve their health status in the short term compared with curative medical services, so they were not willing to actively cooperate with BPHS.
“Some people who had taken physical examination for three years and there was nothing wrong with their body are not willing to accept services anymore, because they believed it was useless for them.”*(D5)*

Third, rural residents who were working/studying outside the village or busily occupied by some physically demanding work thought BPHS was troublesome and interruptive, so they were reluctant to devote any more time and energy to participate in BPHS.
“Many young people migrate to other villages or cities for well-paid work or children’s education. Their parents also live with them outside the original village, and they are reluctant to specially return to their home towns to receive BPHS due to some reasons, such as high transportation cost and busy work, which affects the implementation of BPHS.”*(B6)*

In addition, village doctors affected the cooperation of residents. On the one hand, the limited level of technology and the lack of attention to BPHS work have resulted in low level quality and effectiveness of services, which in turn has led to low levels of cooperation among residents. On the other hand, a monotonous and out-of-date way of propaganda “brochure”, as the only way, was used to publicize and promote BPHS, and the magnitude of disseminating was quite small, which was not enough to arouse villagers’ interest in participating in BPHS.
“The villagers did not pay attention to the follow-up, and the villagers did not follow the life guidance of villager doctors. And My ability is limited, so the appeal is not good.”*(D3)*
“Now in rural areas, propaganda for BPHS is not enough. Despite the handbooks issued, people are unwilling to read them, even some illiterate residents can not read them. People just throw them away, which not only damage propaganda effect, but also causes waste. Now, there are still people who have no idea what BPHS is.”*(C3)*

Village doctors also present some possible suggestions to improve villagers’ cooperation on BPHS.
“Government departments, THCs and village clinics should use a wide variety of modern media, such as Wechat promotion, public-interest advertisements, drama performances and so on. Let the majority of residents understand BPHS, and improve their health awareness, which can guide them to cooperate with us in carrying out these services.”*(B1)*
“People like free things. Sometimes I give people some small gift, such as vitamin pills, eggs, toilet soap, while providing BPHS, to attract people to receive BPHS and gradually increase their health awareness, and the effect is good!”*(C16)*
“Health education should start from children because it will be difficult to correct people’s unhealthy behavior after bad habits are developed.”*(C12)*

## 4. Discussion

The results from this field survey suggested village doctors were facing multitude of challenges, which has posed a severe threat to a high-quality BPHS provision in rural Shandong, China. In summary, the shortage, older, gender imbalance, and poor education/qualification of village doctors; low income; lack of social security; inappropriate performance assessment; inadequate BPHS training; heavy workload; and insufficient cooperation from rural residents have significantly challenged the quality and accessibility of care in BPHS. Many of these obstacles were interlinked and cumulatively contributed to or exacerbated existing difficulties.

### 4.1. Strengthen the Construction of Village Doctor Team

This study demonstrated that the shortage of village doctors was an urgent problem in the current BPHS delivery, which is consistent with other studies from developing countries [35]. According to the China Health Statistics Yearbook, the number of village doctors in Shandong Province has decreased from 142,834 in 2014 to 119,581 in 2017 [36], while the BPHS package has enriched been from 9 categories in 2009 to 14 categories in 2017, which led to an increasing workload on remaining village doctors, damping their enthusiasm and reducing the time for them to provide regular profitable medical services [37]. All of these, in turn, led to a greater reduction in the number of current village doctors [33]. The Chinese government has committed to increasing the retention and recruitment of village doctors through an oriented education program by enacting relevant policy that graduates majoring in medicine are encouraged to work in THCs and village clinics by waiving all their university tuition fees [38]. There is no doubt this program, in the long term, will educate thousands of medical students to go into rural areas and serve people who cannot afford expensive medication. For example, in 2015, a total of 122 of medical college students had been designated to village clinics in Beijing, Shanghai, Xiangyang, and Qingdao [39].

The existing literature has identified that aging of village doctors was also a national issue [40]. However, because this study used purposive sampling, age distribution is not representative. However, we found that the village doctors in some village clinics are older. Older village doctors had limited energy and physical strength to take up the heavy workload of BPHS, and their existing knowledge, especially their raw manipulation on how to use electronic products, has made it difficult for them to meet the requirements of BPHS. For example, the establishment of villagers’ electronic health records, which required using a computer to enter health data, brought new challenges to older village doctors as they are less familiar with modern IT technology. In addition, evidence was found that the gender distribution of village doctors was uneven in this article, with most village doctors being male. Although there is no special requirement for the sex ratio of village doctors in China, the existing literature has identified that in some traditional rural areas, the lack of female village doctors was one of the key barriers to the implementation of BPHS involving women such as maternal and gynecological care [41], because some traditional customs and conventions imbedded in mind often prevented rural women from seeking such care from male village doctors [42].

Information gathered in this research indicated that the quality of BPHS delivery was limited by the poor education and qualifications of the village doctors. In 2002, the Ministry of Health, China, issued the document “Outline for the Development of Health Manpower in China from 2001 to 2015” [43], which required all village doctors to gain a technical secondary school diploma and above, and by 2015, 85% of village clinic staff who gained the certification of Rural Doctor should shift to Practicing (Assistant) Physician in order to better meet the challenges in BPHS provision. However, the results found in this study did not meet the requirements at present. Besides, lack of preventive medicine and public health professionals hinders the improvement of basic public health quality. Traditional Chinese medicine education and Western medicine education include some public health and preventive medicine, but they pay more attention to the treatment after illness and the individual. Through interviews, some village doctors also indicated that medical thinking could not meet the requirements of BPHS in reality. However, preventive medicine and public health specialty focus more on prevention, mainly study the influence of social and environmental factors, and pay more attention to the health of the population [44]. The service object of basic public health is all residents, and the purpose is to improve the health level of all residents, which requires health providers to have a great view of health, to change the concept of emphasizing treatment over prevention. In other words, carrying out work with medical thinking alone cannot better meet the needs of BPHS work. Therefore, it is necessary to introduce and train more public health professionals.

Moreover, unprofessional and inadequate BPHS training for villager doctors deteriorated the quality of BPHS delivery to some extent. Timely and comprehensive training programs initiated for village doctors serve a significant function in increasing the effectiveness of BPHS delivery and satisfaction of rural residents by updating village doctors’ knowledge and skill, enhancing their awareness of the importance of BPHS in promoting health and preventing diseases instead of treating the illness, and lighting up their sense of value and mission on what they are doing. Also, because of the illiteracy of information technology of some elderly village doctors, the BPHS training should be tailored to meet their needs so as to improve their service capability. For example, it could be much easier for them to create electronic health records on computer, if one or two local villagers who developed a good command with computers can be hired to help them [33].

### 4.2. Improve Income and Social Security of Village Doctors

This study, as well as previous literature [22,45], manifested that low income and lack of social security have a detrimental influence on village doctors’ motivation and passion on providing high quality BPHS. This was also one of the important reasons why it is arduous to attract new qualified village doctors to serve in rural areas [46]. Other research even cited low salary as the main reason why Chinese primary health workers switched careers [8]. To retain primary healthcare workers, the Australian government, using a feasible incentive plan, financially supported the healthcare workforce according to the classification of geographical location, in order to ensure healthcare workers in rural areas, especially in remote areas, receive higher income and satisfying employee benefits [47]. Recently, the Chinese government has been actively taking measures to solve this unsatisfactory situation. Premier Li Keqiang, from the government angle, proposed that “Try our best to make village doctors really attractive occupation. Let the competent village doctors be willing to stay in the countryside by improving their treatment in a variety of ways. Relieve the worries of village doctors by safeguarding their reasonable income and improving their pension benefits” [48]. Another policy document “Guidance on further strengthening the construction of village doctors team”, issued by the State Council of China, clearly stated that a wide range of measures should be taken to provide robust social security for village doctors [49]. For example, some areas like Guangdong, Henan, and Anhui have offered an extra bonus to village doctors according to the amount of working years. Jiangsu Province and Zhejiang Province also keep the pace by paying a certain percentage of premium to help rural doctors be enrolled in social security such as pension insurance [50]. Li’s research delivered village doctors’ voice that they were not afraid of the heavy workload of BPHS, as long as they can be well rewarded and feel a sense of pride and belonging [6]. Therefore, improving the income and social security of villager doctors plays significant role in the long-term high-quality delivery of BPHS in China’s rural areas. It is necessary to quantitatively assess the relationship between the amount of BPHS subsidies, the social security, and the willingness and enthusiasm of rural doctors to provide BPHS [22].

### 4.3. Improve the Performance Assessment Method

Unreasonable performance assessment on village doctors was an obvious obstacle for providing good BPHS in many rural areas [51]. Evidence found in this article illustrates that despite the establishment of supervision and assessment mechanism on village doctors’ BPHS delivery, there were still problems identified in the implementation process. Performance assessment did not appear to be as effective as it should be [22]. First of all, subtle distinction in BPHS subsidies issued to village doctors based on the assessment result undermined initiatives of village doctors. Secondly, unrealistic and formalistic assessment criteria have been gradually deteriorating the quality of BPHS provision. For example, for one thing, the telephone return visit to villagers has aroused the discontent of most village doctors, because THCs do not always take some special circumstances into account, like the fact that some elderly people missed the call or they just forgot the services provided by village doctors for the time being because of their degraded memory, which greatly influenced performance assessment on villager doctors; for another thing, emphasizing quantity but neglecting quality during assessment has led to village doctors attaching more importance on the quantity of BPHS provision instead of high quality [52]. Thirdly, no feedback responds to village doctors from THCs after the assessment, which was not conducive to the improvement of the future BPHS work of village doctors. Therefore, THCs should employ more multidimensional, more practical, and more flexible measures to assess the BPHS delivery of village doctors in order to guarantee fairness and veracity, and reduce rural healthcare workers’ dissatisfaction. To be specific, one possible way is that THCs can organize an assessment group including village doctors as representatives, then not only can mutual supervision be developed, but also village doctors can gain a better and deeper understanding about the assessment criteria during the performance assessment simultaneously. Moreover, it is necessary for THCs to disclose the scoring rules of the performance assessment, strengthen the link between assessment results and subsidies, and give timely feedback so as to assist village doctors to learn from each other’s strengths and weaknesses. Furthermore, the assessment should be gradually shifted from quantity-based to quality-based or result-based by reasonably assigning tasks, and optimizing the assessment process and frequency [6]. It may be a beneficial strategy in rural China to learn from the United States’ and Australian practice of developing various qualitative indicators when conducting performance assessment on village doctors for improving their service quality [53].

### 4.4. Facilitate Cooperation among Villagers

The present study, as well as other studies, showed that insufficient cooperation among villagers was an important factor hindering the high-quality delivery of rural BPHS [12]. Because of the insufficient number and quality of village doctors, it is not enough to provide and popularize the high quality of BPHS, so that the villagers’ cooperation is at a low level, so that the villagers’ cooperation is not enough. At the same time, the villagers themselves have some problems that have largely hindered the village doctors from providing basic public health services. Several measures may be taken to improve villagers’ understanding and cooperation on BPHS. First, a wide variety of propaganda methods can be employed to conduct health education and raise villagers’ health awareness in order to better understand and appreciate BPHS in daily life, such as public-interest advertising on TV, wireless broadcasting, brochures or flyers, social media on the Internet, and so on, instead of using monotonous methods. Second, distributing some free small gifts to villagers, like eggs, vitamin, towel, and soap, could be a practical way to attract them to actively coordinate with BPHS. During our investigation, we know that Qingdao city has used this method to facilitate cooperation among villagers and achieved good results. Third, just as a Chinese saying goes, “Start with children”, BPHS education programs should be held in schools, which could be a long-sighted plan to improve people’s health awareness and improve their cooperation with BPHS in the future.

## 5. Conclusions

At present, village doctors in Shandong province are facing a wide range of challenges in providing higher-quality BPHS, including insufficient number and quality of personnel, low income and lack of security, unreasonable performance appraisal system, and low level of villagers’ cooperation. Thus, it is urgent for government officials and policy makers to develop appropriate and practical strategies to overcome these challenges, such as increasing the income and security of village doctors to attract new health providers to work in village clinics, sharpening the professional skills of existing village doctors through strengthening technical training, establishing fairer and more effective performance appraisal system, and strengthening publicity and health education for villagers. Moreover, the findings of the present study may also be helpful for other parts of China and other developing countries in recognizing possible rural area health provision shortfalls and developing effective strategies to improve the quality of primary health services.

### Limitations

Several limitations of the present study should be taken into account. First, villagers, staff of THCs and CDC, and relevant government officials who might also contribute to the identification of challenges faced by village doctors in provision of BPHS were not interviewed in this research. Second, the interviewees were only selected from rural Shandong. Because of different socio-economic development and demographic characteristics, the evidence found in this article may not be totally generalized to other areas in China. Therefore, this study can be viewed only as preliminary. Further studies need to be conducted by engaging more respondents and in different areas in order to gain deeper understanding of this topic.

## Figures and Tables

**Table 1 ijerph-16-02519-t001:** Village doctors’ demographics (*n* = 129).

Characteristic	Number	%
**Age**		
≤35	13	10.08
36–45	60	46.51
46–55	44	34.11
≥56	12	9.30
**Gender**		
Male	83	64.34
Female	46	35.66
**Education**		
Bachelor degree and above	1	0.78
Junior college	27	20.93
Secondary school	99	76.74
Senior high school	1	0.78
Junior high school and below	1	0.78
**Specialty**		
Chinese Traditional Medicine	11	8.53
Western Medicine	73	57.36
Integrated of Traditional and Western Medicine	37	28.68
Others ^1^	7	5.43
**Qualification**		
Practicing Physician ^2^	13	10.08
Practicing Assistant Physician ^3^	14	10.85
Rural Doctor ^4^	100	77.52
Disqualified	2	1.55
**Years of starting work as a village doctor**		
≤5	14	10.85
6–10	21	16.28
11–15	11	8.53
16–20	24	18.60
≥21	57	44.19
Unknown	2	1.55

^1^ people who majored in non-medical specialty; ^2,3^ a kind of qualification certificate granted by Ministry of Health to village doctors who has passed the national unified medical examination and gained a certificate with the level of “Practicing Physician” or “Practicing Assistant Physician”; ^4^ a practice certificate issued by the health administrative department of the local county-level government to a person who passed the corresponding registration and training examination [12].

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
