# Peer review of "What Are the Challenges Faced by Village Doctors in Provision of Basic Public Health Services in Shandong, China? A Qualitative Study"

_ijerph, 2019, doi:10.3390/ijerph16142519_

Round 1
Reviewer 1 Report
This paper sheds light on the challenges faced by village doctors in provision of basic public health services in China. The introduction is well written.
My concerns are as follows:
1. Data: The authors report the challenges faced by village doctors, but 7 nurses are analyzed in table 1.
2. Results: The authors refer to “Insufficient cooperation from villagers”. In fact, the reasons for insufficient cooperation of villagers include various aspects such as the doctors. But the results only describe the reasons from villagers.
3. Conclusions: It’s not very clear. The authors should make conclusion according to the results and discussion of this paper.
Author Response
Dear Reviewers,
Thanks very much for the comments, which are very helpful for us to improve the manuscript. Now we have submitted a revised version of the manuscript. All concerns have been addressed one-by-one as follows. Hopefully it meets the requirement of acceptance for publication.
COMMENT: Data: The authors report the challenges faced by village doctors, but 7 nurses are analyzed in table 1
RESPONSE: To avoid ambiguity, we removed 7 registered nurses, rearranged the data (Table 1) and made adjustments in the results and discussion section. However, in the survey, there are indeed nurses in the few clinics where there are two or more people, and some of them have obtained the nurse qualification certificate.
COMMENT: Results: The authors refer to “Insufficient cooperation from villagers”. In fact, the reasons for insufficient cooperation of villagers include various aspects such as the doctors. But the results only describe the reasons from villagers.
RESPONSE:
We have added the sentence in result as follows,
“In addition, village doctors affected the cooperation of residents. On the one hand, the limited level of technology and the lack of attention to BPHS work have resulted in low level quality and effectiveness of services, which in turn has led to low levels of cooperation among residents. On the other hand, a monotonous and out-of-dated way of propaganda “brochure”, as the only way, was used to publicize and promote BPHS, and the magnitude of disseminating was quite small, which was not enough to arouse villagers' interest in participating in BPHS.”
“The villagers did not pay attention to the follow-up, and the villagers did not follow the life guidance of villager doctors. And My ability is limited, so the appeal is not good.”(D3)[ Line 556-563]
Corresponding, we have added the sentence in discussion as follows,
“Due to the insufficient number and quality of village doctors, it is not enough to provide and popularize high quality of BPHS, so that the villagers' cooperation is in low level. At the same time, the villagers themselves have some problems that have largely hindered the village doctors from providing basic public health services.”[ Line 727-731]
COMMENT: Conclusions: It’s not very clear. The authors should make conclusion according to the results and discussion of this paper.
RESPONSE: We have rephrased this sentence in Conclusion as follows,
“At present,village doctors in Shandong province are facing a wide range of challenges in providing higer quality BPHS, including insufficient number and quality of personnel, low income and lack of security, unreasonable performance appraisal system, and low level of villagers' cooperation. Thus,it is urgent for government officials and policy makers to develop appropriate and practical strategies to overcome these challenges, such as increasing the income and security of village doctors to attract new health providers to work in village clinics, sharping the professional skills of existing village doctors through strengthening technical training, establishing fairer and more effective performance appraisal system, and strengthening publicity and health education for villagers. Moreover, the findings of present study may also be helpful for other parts of China and other developing countries in recognizing possible rural area health provision shortfalls and developing effective strategies to to improve the quality of primary health services.” [Line 743-753]
Reviewer 2 Report
The topic of the paper is very interesting, however the research is based on very simple methods and does not bring much from the scientific point of view.
The narrative part of the manuscript, in the introduction, as well as in the described results and the discussion part are rather general, devoid of deeper insights.
I would suggest to give a more detailed description of why this research area was chosen, what important features it was characterized by. The same applies to the selected sample of respondents.
There are also a few issues that can confuse potential readers.
Line 156 and further the subsection: 3.1. Village clinic staffing distribution
According to the authors, the number of the village doctors’ (“participants”) was 136. I am a bit confused studying Table 1 and the given data (as well as the explanation in the text). How is it possible that all of them are called “village doctors”, if for example 1 of them has only “junior high school and below education”, 1 of them “senior high school education”, 101 “secondary school” education? Why they are called “village doctors” if they are not really doctors (MDs, with a university degree)? Only 2 of the “participants” have a “bachelor degree and above”. Is it possible to be a doctor without a university diploma? I’m a bit confused.
I would like to ask the authors to explain / define their use of the “village doctor” word.
Do the authors examined village doctors (village MDs), or they rather examined health professionals in general and have included in their research also medical rescuers, paramedic, nurses, other medical professionals, etc.
To avoid such misunderstandings, the professional position of the respondents (“participants”) should be precisely defined.
Some statements / conclusions put forward by the authors can be considered as debatable, e.g. lines 189-197:
I have some doubts whether the authors can conclude (based on the results of their own research) that a challenge / problem of rural doctors is ageing? This is a very controversial issue. According the findings reported by the authors 45.59% of the “participants” were 36-45 years old and 33.09% were 46-55 years old. It can be considered that they are not old (especially the group between 36 and 45). They could be rather considered as experienced doctors with practical knowledge and extensive experience. This requires a broader discussion.
The part 5.Conclusion is very general, does not relate sufficiently to the obtained study results.
Author Response
Dear Reviewers,
Thanks very much for the comments, which are very helpful for us to improve the manuscript. Now we have submitted a revised version of the manuscript. All concerns have been addressed one-by-one as follows. Hopefully it meets the requirement of acceptance for publication.
Reviewer2
COMMENT: I would suggest to give a more detailed description of why this research area was chosen, what important features it was characterized by. The same applies to the selected sample of respondents.
RESPONSE:
“……studies on this subject are still lacking in Shandong which has the largest number of village doctors (113,465)in 2016, ranking first in the country. It is worth noting that Shandong has a large rural population ranking second in China……” [Line88-90]
As for why we selected sample of village doctors, because “……supplying high quality of BPHS in rural China is of great significance in improving population’s quality of life and promoting social harmony. Currently, about 1 million village doctors undertake approximately 40% of the BPHS workload, which also lays stress on village doctors’ significant function in delivering BPHS to rural residents, so the quality of BPHS provided by village doctors has a direct influence on the health status of rural residents. However, few researches has uncovered village doctors are facing many challenges in BPHS provision……”[Line59-67]
COMMENT:
There are also a few issues that can confuse potential readers. Line 156 and further the subsection: 3.1. Village clinic staffing distribution
According to the authors, the number of the village doctors’ (“participants”) was 136. I am a bit confused studying Table 1 and the given data (as well as the explanation in the text). How is it possible that all of them are called “village doctors”, if for example 1 of them has only “junior high school and below education”, 1 of them “senior high school education”, 101 “secondary school” education? Why they are called “village doctors” if they are not really doctors (MDs, with a university degree)? Only 2 of the “participants” have a “bachelor degree and above”. Is it possible to be a doctor without a university diploma? I’m a bit confused.
I would like to ask the authors to explain / define their use of the “village doctor” word.
RESPONSE:
The title of the table1 "participant demographics " has been changed to "village doctors demographics". [Line 219]
As for the problem of “Is it possible to be a doctor without a university diploma”, this is related to China’s national conditions.
In China, village doctors were formerly barefoot doctors. Barefoot doctors refer to rural medical personnel who have not received formal medical training but still hold agricultural hukou and are "half farmer and half doctor" in some cases. There were three main sources of village doctors at that time: first, medical families; Second, high school graduates who slightly understand pathology; Three is some educated young people who decided to work in the countryside. In 1960s, the Chinese government began to implement the training program for rural health workers and established a team of front-line health service providers.
In the early 1980s, when China implement opening up policy and economic reforms, barefoot doctors were unable to meet the growing demand for health services of rural population. In 1985, Chinese government stopped using the term "barefoot doctor" and replaced it with "village doctor". Barefoot doctors need to pass an exam to get village doctor's certificate. Those who fail will quit the health service industry. The “Regulations on the Administration of Village Doctors” implemented since January 1, 2004, stipulates that village doctors, after passing the corresponding registration and training examination and obtain the practice certificate, can then start business with the formal license6. Village doctors are encouraged to apply for the standardized certificate like practicing physician certificate. The initial training level for village doctors was only a few months, but now they must receive two to three years of professional training, which is equivalent to the high school diploma. [Line 43-57]
Therefore, due to China's special national conditions, there are few people with a bachelor's degree or above among village doctors. After receiving secondary vocational education, most of them begin to provide medical services in the village and then conduct on-the-job training.
COMMENT: Do the authors examined village doctors (village MDs), or they rather examined health professionals in general and have included in their research also medical rescuers, paramedic, nurses, other medical professionals, etc.
To avoid such misunderstandings, the professional position of the respondents (“participants”) should be precisely defined.
RESPONSE: The definition was given in page 3. “Village doctors refer to those who provide basic medical service and BPHS in village clinics,except nurses and other medical professionals. ”[Line 137-138]
COMMENT: Some statements / conclusions put forward by the authors can be considered as debatable, e.g. lines 189-197: I have some doubts whether the authors can conclude (based on the results of their own research) that a challenge / problem of rural doctors is ageing? This is a very controversial issue. According the findings reported by the authors 45.59% of the “participants” were 36-45 years old and 33.09% were 46-55 years old. It can be considered that they are not old (especially the group between 36 and 45). They could be rather considered as experienced doctors with practical knowledge and extensive experience. This requires a broader discussion.
RESPONSE: We have rephrased this sentence in 3.2.2 as follows,
3.2.2. Older Village doctors in some villages
Age of village doctors in some villages may be another challenge to BPHS. In some of the clinics, village doctors are older, and the service population is large, which greatly reduced the quality of service. For example, in village C3 village, there are two village doctors in the clinic, one 55 years old and one 58 years old, serving 1609 villagers. There are even village B1 where population is 1,800, and there is only one village doctorwho is 57 years old in the village clinic. Older age has become another major obstacle for village doctors in providing high-quality BPHS. Older village doctors often needed more time and energy to learn and comprehend new knowledge about how to provide better BPHS. A it is also difficult for them to develop a good command of new skills which are necessary for delievering BPHS, such computer proficiency due to the deterioration of physical function and memory. Another worrying situation is that some older villages faced the problem of having no successor after the retirement. [Line302-313]
Correspondingly, the discussion section was revised as follows,
The exisiting literature has identified aging of village doctors was also a national issue [39]. But because this study used purposive sampling,age distribution is not representative. However, we found that the village doctors in some village clinics are older. [Line605-607]
COMMENT: The part 5.Conclusion is very general, does not relate sufficiently to the obtained study results.
RESPONSE: We have rephrased this sentence in Conclusion as follows,
“At present,village doctors in Shandong province are facing a wide range of challenges in providing higher quality BPHS, including insufficient number and quality of personnel, low income and lack of security, unreasonable performance appraisal system, and low level of villagers' cooperation. Thus,it is urgent for government officials and policy makers to develop appropriate and practical strategies to overcome these challenges, such as increasing the income and security of village doctors to attract new health providers to work in village clinics, sharping the professional skills of existing village doctors through strengthening technical training, establishing fairer and more effective performance appraisal system, and strengthening publicity and health education for villagers. Moreover, the findings of present study may also be helpful for other parts of China and other developing countries in recognizing possible rural area health provision shortfalls and developing effective strategies to to improve the quality of primary health services.”[Line 743-753]
Reviewer 3 Report
This is a very important topic and your extensive work interviewing and analyzing is impressive. In general, your manuscript is well done. Attached is a copy of the document with recommended edits primarily related to analysis.

Author Response
Dear Reviewers,
Thanks very much for the comments, which are very helpful for us to improve the manuscript. Now we have submitted a revised version of the manuscript. All concerns have been addressed one-by-one as follows. Hopefully it meets the requirement of acceptance for publication.
COMMENT: How were the observations factored into analysis?
“During the interviews, body language of the interviewees was carefully observed and noted.”
RESPONSE: During the interviews, body language, such as facial expression, tone and small movements of the interviewees was carefully observed and noted, so as to help judge the authenticity of the interviewee's words. If there is any problem, the interviewer will continue to ask questions to collect more authentic information.[ Line 155-159]
COMMENT: It would be helpful to define village doctor vs. practicing physician for readers unfamiliar with the terms.
RESPONSE: A brief introduction of village doctor vs. practicing physician has been presented in the article.
2,3 a kind of qualification certificate granted by Ministry of Health to village doctors who has passed the national unified medical examination and gained a certificate with the level of “Practicing Physician” or ” Practicing Assistant Physician “
4 a practice certificate issued by the health administrative department of the local county-level government to a person who passed passing the corresponding registration and training examination. [ Line 239-243]
COMMENT:Consider age issues more holistically.
RESPONSE: We have rephrased this result as follows,
3.2.2. Older Village doctors in some villages
Age of village doctors in some villages may be another challenge to BPHS. In some of the clinics, village doctors are older, and the service population is large, which greatly reduced the quality of service. For example, in village C3 village, there are two village doctors in the clinic, one 55 years old and one 58 years old, serving 1609 villagers. There are even village B1 where population is 1,800, and there is only one village doctorwho is 57 years old in the village clinic. Older age has become another major obstacle for village doctors in providing high-quality BPHS. Older village doctors often needed more time and energy to learn and comprehend new knowledge about how to provide better BPHS. And it is also difficult for them to develop a good command of new skills which are necessary for delievering BPHS, such computer proficiency due to the deterioration of physical function and memory. Another worrying situation is that some older villages faced the problem of having no successor after the retirement. [Line302-313]
COMMENT: Public health and preventive medicine are important. Are these educational components included or absent in western medicine education? Other education? It is important to consider which type of education includes preventive and public health information in order to fully and completely analyze what is missing from the education of various types of providers.
RESPONSE: Besides, lack of preventive medicine and public health professionals hinder the improvement of basic public health quality. Traditional Chinese medicine education and western medicine education include some public health and preventive medicine, but they pay more attention to the treatment after illness and the individual. Through interviews, some village doctors also indicated that with medical thinking could not meet the requirements of BPHS in reality. However, preventive medicine and public health specialty focuses more on prevention, mainly studies the influence of social and environmental factors, and pays more attention to the health of the population. The service object of basic public health is all residents, and the purpose is to improve the health level of all residents, which requires health providers to have a great view of health, to change the concept of emphasizing treatment over prevention. In other words, carrying out work with medical thinking alone cannot better meet the needs of BPHS work. Therefore, it is necessary to introduce and train more public health professionals. [Line625-637]
COMMENT: So, reimbursement and benefits models must be updated to align with role expectations?
RESPONSE: We have responded to reviewer’s comment by presenting
“It is necessary to quantitatively assess the relationship between the amount of BPHS subsidies, the social security and the willingness and enthusiasm of rural doctors to provide BPHS”. [Line 680-682]
COMMENT: To address this reality, some health care systems (US and elsewhere)have shifted to a pay for performance system.
RESPONSE: A performance evaluation system has been established in the survey area. In principle, basic public health subsidies should be issued according to the performance evaluation results, but there will be some problems during the specific implementation. Although the scores will be given during the assessment, the differences of subsidies issued are subtle and the incentive effect is not exerted.
COMMENT: This is a real not a potential limitation. Further research that takes various perspectives into consideration is important.
RESPONSE: We have rephrased this sentence as follows,
“Several limitations of the present study should be taken into account.” [Line 755]
Round 2
Reviewer 1 Report
The paper has some minor errors, and could benefit from a proofreading. For example,
on page 4, line 176 and 180, “village doctors health workers”; on page 14, line 609, “effective strategies to to improve the quality of primary health services”.
Reviewer 2 Report
Thank you for making the necessary corrections and giving explanations that made the results of the research, and thus the article more understandable.